# Impact of digital health interventions on pain and symptom management in home hospice patients: A systematic review and meta-analysis protocol

Thiago Oliveira Dos Santos[1], Fábila Fernanda Dos Passos Da Rosa[1],
Kleyton Santos Medeiros[2], Francis Solange Vieira Tourinho[1]*

1 Federal University of Santa Catarina, Florianópolis, Santa Catarina, Brazil, 2 Instituto de Ensino, Pesquisa e Inovação. Liga Contra o Câncer, Natal, Brazil

* francis.tourinho@ufsc.br

## Abstract

### Background

The growing global demand for palliative care, particularly in low- and middle-income countries, underscores the urgent need for innovative strategies to ensure symptom control and quality of life for patients with advanced illnesses. Among these strategies, digital health interventions (DHIs) have emerged as promising tools to support pain and symptom management in home hospice settings, especially for cancer patients.

### Objective

This protocol outlines a systematic review and meta-analysis designed to evaluate the effectiveness of digital health interventions in managing pain and other symptoms among cancer patients receiving palliative care at home.

### Methods

This review will include randomized clinical trials and quasi-randomized clinical trials that assess the impact of digital health interventions on pain and symptom control in adult cancer patients under palliative care at home. Studies will be identified through searches in major bibliographic databases, following PRISMA-P guidelines and the Cochrane Handbook. Only peer-reviewed publications will be considered. Interventions must align with the World Health Organization's definition of DHIs. The primary outcome is the effectiveness of these interventions in pain management compared to traditional in-person care.

**Data availability statement:** N/A.

**Funding:** The author(s) received no specific funding for this work.

**Competing interests:** The authors have declared that no competing interests exist.

## Expected results

This review is expected to identify and synthesize evidence on the effectiveness of DHIs in improving pain and symptom control, as well as overall patient well-being in palliative home care settings,

## Conclusion

The findings will provide a comprehensive understanding of the role of digital technologies in enhancing palliative care delivery, guiding clinical practices and policy decisions aimed at optimizing end-of-life care through remote support. This protocol is registered in the International Prospective Register of Systematic Reviews (PROSPERO): CRD420250572413

## Introduction

Currently, estimated 56.8 million people need palliative care, most of whom live in low- and middle-income countries, with approximately 34% of these patients diagnosed with some form of cancer. Furthermore, as life expectancy increases and populations age, particularly in developed countries, the global demand for palliative care among adults is expected to rise significantly in the coming years (WHO, 2020).

Globally, only one in ten individuals in need of palliative care, defined as treatment aimed at alleviating pain, symptoms, and psychological distress associated with serious and chronic illnesses, receive such care [1].

It is important to highlight that, although most PC is provided in high-income countries, almost 80% of the global need is in low- and middle-income countries. Regrettably, only 20 countries have PC well integrated into their health-care systems [2].

Most patients requiring palliative care express a preference for receiving care in their homes, and many can do so while obtaining the necessary support and medical attention [3]. Among individuals with end-stage cancer, the desire to receive care and die at home is common in Western countries [4].

Patients often express a desire to find meaning and maintain a sense of control over their lives; consequently, they prefer having autonomous access to healthcare professionals when needed. Furthermore, individuals receiving palliative care value continuity of care and favor an integrated approach in which healthcare providers deliver coordinated and comprehensive support [5].

Home is the most preferred place of end-of-life care for both patients (11%–89%) and family members (23%–84%) [6]. It is also the most preferred place of death, with patient estimates ranging from 51% to 55% [6].

In this context, telemedicine has emerged as a tool that ensures continuity of clinical support and effective management of signs and symptoms, such as pain, even in remote areas, particularly for cancer patients receiving palliative care [7].

It is important to note that most cancer patients experience a range of symptoms, the prevalence and severity of which vary depending on the type and stage of cancer, the treatments received, and the presence of comorbidities. In advanced cancer, 35%

to 96% of patients experience pain, 32% to 90% experience fatigue, and 10% to 70% experience breathlessness. Patients typically experience more than one symptom at any one time [8].

Despite advances in the field, delivering effective palliative care remains challenging, particularly in remote or resource-constrained settings [9,10]. The advancement of smart technologies in healthcare presents promising opportunities to address these barriers [11]. However, the application of such technologies within palliative care is still an emerging area of study, with many aspects yet to be thoroughly investigated [12].

In this context, the integration of palliative care and telemedicine has proven to be an effective approach for enhancing the quality of life of patients with serious illnesses, including those nearing the end of life. Telehealth serves as a critical strategy to ensure continuity of care focused on the management of pain and other distressing symptoms [13].

In this context, the integration of palliative care and telemedicine has proven to be an effective approach for enhancing the quality of life of patients with serious illnesses, including those nearing the end of life. Telehealth serves as a critical strategy to ensure continuity of care focused on the management of pain and other distressing symptoms [13]. Thus, 53.5% of palliative care healthcare professionals believe that virtual consultations have the potential to replace face-to-face interactions at a similar level [14].

Although the use of technology in palliative care for adult oncology patients holds considerable promise for improving quality of life, significant gaps remain in the literature, particularly regarding the effectiveness and acceptance of these tools in specific contexts, such as remote pain management compared to traditional in-person outpatient care. Therefore, this study aims to evaluate the effectiveness of such interventions, with a focus on pain management and enhancing the quality of life for these patients.

### Review question

What is the effectiveness of digital health interventions in managing pain among cancer patients receiving palliative care, compared to traditional in-person care?

### Aim

To evaluate the effectiveness of digital health interventions in managing pain and other symptoms among cancer patients receiving palliative care in home hospice settings.

## Materials and methods

This systematic review and meta-analysis will be reported in accordance with the Preferred Reporting Items for Systematic Reviews and Meta-Analyses Protocols (PRISMA-P) guidelines [15]. Additionally, the protocol was developed based on the recommendations outlined in the Cochrane Handbook for Systematic Reviews of Interventions [16]. This protocol has been registered in the International Prospective Register of Systematic Reviews (PROSPERO).

### Ethical considerations

This study is a systematic review protocol, and the research will be conducted through bibliographic database searches without the involvement of patient data. As such, the research questions do not involve human participants. Similarly, outcome assessments during the design, implementation, and dissemination of the study will not include any patient data. Thus, secondary data were used in this study, so obtaining approval from the ethics committee was not necessary.

### Inclusion criteria

Randomized clinical trials and quasi-randomized clinical trials evaluating the digital health interventions in managing pain and other symptoms among cancer patients receiving palliative care in home hospice settings will be included in the sample. Only studies involving adult patients aged 18 years and older will be included.

Digital health interventions (DHIs) will be considered as technology-based solutions aimed at improving health outcomes, supporting healthcare systems, and enhancing patient engagement, in accordance with the definition provided by the World Health Organization [17].

### Exclusion criteria

Published but not peer-reviewed articles will be excluded from the review. Observational studies, review articles, reports, and case series will also be excluded. Studies evaluating the use of DHIs in hospitals and clinics.

### Patient, intervention, comparison, outcome, and time (PICOT) strategy

The PICOT strategy will be applied as follows:

### Intervention

- Population: oncological patients in palliative care.
- Intervention/Exposure: Digital health interventions (DHIs) (Table 1).
- Comparator/Control: In-person outpatient care.
- Outcomes: pain management and other symptoms.
- Types of studies to be included: randomized controlled trials and quasi-randomized controlled trials.

### Primary outcome

Pain management.

### Secondary outcomes

The secondary outcomes will be anxiety and quality of life.

The assessment of anxiety symptoms will be carried out using the Hospital Anxiety and Depression Scale (HADS) and the Generalized Anxiety Disorder 7-item (GAD-7), both widely used and validated in clinical practice and research. The quality of life of patients will be measured using the Caregiver Quality of Life Index–Cancer (CQOL-C), which considers

**Table 1. Digital health interventions.**

| | |
|---|---|
| Telemedicine | Delivery of health services via remote telecommunications. This includes interactive consultative and diagnostic services [18]. |
| Digital Health | Use of digital technologies in medicine and other health professions to manage illnesses and health risks and to promote wellness; includes the use of wearable devices; health information technology; electronic health records; telemedicine; and personalized medicine [19]. |
| Mobile Applications | Computer programs or software installed on mobile electronic devices which support a wide range of functions and uses which include television, telephone, video, music, word processing, and Internet servic [20]. |
| Remote Patient Monitoring | A type of TELEMEDICINE in which healthcare providers monitor patients using digital medical devices. Data collected from these devices are then electronically transferred to providers for care management and has been used to measure symptoms of chronic conditions, such as cardiac diseases, diabetes, and asthma. Benefits include patient engagement, patient adherence to their treatment plan, and the ability to expand physician reach and easily provide care to patients without the need for patients in person encounters [21]. |

the physical, emotional, social, and functional aspects of the individual. To measure pain, the Palliative Care Initial Screening (PINS) pain scale and the Numeric Rating Scale (NRS) will be used, allowing for both objective and subjective evaluations of symptom intensity [22–24].

### Search strategy

The following databases will be used **(S1 File: Peer Review of Electronic Search Strategies):** PubMed, ScienceDirect, Embase, CINAHL, LILACS, CENTRAL, Web of Science, Scopus e Cochrane Library. No language or publication period restrictions will be imposed. The Medical Subject Headings (MeSH) terms will be (Table 2):

### Other sources

The reference lists of the retrieved papers may also be used to choose appropriate research. In other words, the reference lists of articles that were retrieved may allow the computerized literature search to be expanded. Identical strategies will be applied to other databases.

### Data collection and analysis

**Study selection.** Following the database searches, all retrieved records will be imported into Rayyan software, where duplicate entries will be removed. Two reviewers (TOS and KSM) will independently screen titles and abstracts based on

Table 2. Search strategy.

| Medline/PubMed | (((("Palliative Care"[Mesh] OR (Care, Palliative) OR (Palliative Supportive Care) OR (Supportive Care, Palliative) OR (Palliative Treatment) OR (Palliative Treatments) OR (Treatment, Palliative) OR (Treatments, Palliative) OR (Palliative Therapy) OR (Therapy, Palliative) OR (Palliative Surgery) OR (Surgery, Palliative)) AND ((("Telemedicine"[Mesh] OR (Virtual Medicine) OR (Medicine, Virtual) OR (Tele-Referral) OR (Tele Referral) OR (Tele-Referrals) OR (Mobile Health) OR (Health, Mobile) OR (mHealth) OR (Telehealth) OR (eHealth) OR (Tele-Intensive Care) OR (Tele Intensive Care) OR (Tele-ICU) OR (Tele ICU) OR (Telecare) OR (Tele-Care) OR (Tele Care) OR ("Digital Health"[Mesh] OR (Health, Digital) OR (Digital Health Technology) OR (Digital Health Technologies) OR (Health Technologies, Digital) OR (Health Technology, Digital))) AND ("Pain Management"[Mesh] OR (Management, Pain) OR (Managements, Pain) OR (Pain Managements)) |
|---|---|
| Web of Science | ("Pain" OR "Pains" OR "Physical Suffering" OR "Physical Sufferings" OR "Ache" OR "Aches") AND ("Palliative Care" OR "Palliative") AND ("Telemedicine" OR "Connected Health" OR "eHealth" OR "Health 2.0" OR "Health Tele-Services" OR "Health Teleservices" OR "Medicine 2.0" OR "mHealth" OR "Mobile Health" OR "Tele Care" OR "Tele-Care" OR "Telecare" OR "Telecure" OR "Telehealth" OR "Teleservices in the Health Sector" OR "Virtual Medicine") |
| BVS/LILACS | ("Pain" OR "Pains" OR "Physical Suffering" OR "Physical Sufferings" OR "Ache" OR "Aches" OR "Dor" OR "Algia" OR "Sensação de Ardência" OR "Sofrimento Físico" OR "Dolor" OR "sufrimiento físico") AND ("Palliative Care" OR "Palliative" OR "Cuidados Paliativos" OR Paliativ*) AND ("Telemedicine" OR "Connected Health" OR "eHealth" OR "Health 2.0" OR "Health Tele-Services" OR "Health Teleservices" OR "Medicine 2.0" OR "mHealth" OR "Mobile Health" OR "Tele Care" OR "Tele-Care" OR "Telecare" OR "Telecure" OR "Telehealth" OR "Teleservices in the Health Sector" OR "Virtual Medicine" OR "Telemedicina" OR "Ciber Saúde" OR "Ciber-Saúde" OR "Cibersaúde" OR "e-Saúde" OR "eSaúde" OR "Medicina 2.0" OR "Medicina Virtual" OR "mSaúde" OR "Saúde 2.0" OR "Saúde Conectada" OR "Saúde Eletrônica" OR "Saúde Móvel" OR "Telessaúde" OR "Tele-Serviços em Saúde" OR "Teleassistência" OR "Teles-serviços de Saúde" OR "Telesserviços em Saúde" OR "Telesserviços na Saúde" OR "eSalud" OR "Ciber Salud" OR "Ciber-Salud" OR "Cibersalud" OR "mSalud" OR "Salud 2.0" OR "Salud Conectada" OR "Salud Electrónica" OR "Salud Móvil" OR "Salud Mue-ble" OR "Telesalud" OR "Teleasistencia" OR "Telesalud" OR "Teleservicios de Salud") |

predefined inclusion criteria. Full-text articles of studies deemed potentially relevant will also be assessed independently by the same reviewers. Only studies agreed upon by both reviewers will be included in the final systematic review. In cases of disagreement, a third reviewer (FSVT) will resolve any conflicts. A detailed log of excluded studies, along with justifications, will be maintained throughout the selection process. The study selection procedure and results will be documented using the PRISMA flow diagram (Fig 1).

**Identification of studies via databases and registers**

**Identification**

Records identified from*:
  PUBMED (n=  );
  ScienceDirect (n=  );
  EMBASE (n=  ); CINAHAL
  (n=  ); LILACS (n=  );
  CENTRAL (n=  );  Web of
  Science (n=  ); Scopus (n=
  ); Cochrane (n=  ).

→ Records removed *before screening*:
Duplicate records removed  (n=  )
Records removed for other reasons (n=  )

**Screening**

Records screened (n=   )

→ Records excluded**
Title and abstracts irrelevant to the topic (n=   )

Reports sought for retrieval (n=   )

→ Reports not retrieved (n=   )

Reports assessed for eligibility (n =  )

→ Reports excluded:
Case reports: (n=   )
Publications that are not specifically about Digital Health Interventions on Pain and Symptom Management in Home Hospice Patients: (n=   )
Insufficient data to be extracted or calculated: (n=   )

**Included**

Studies included in review (n=  )
Reports of included studies (n=  )

**Fig 1.  PRISMA flow diagram.**

## Data extraction

A standardized data extraction form will be created and pilot-tested (see S1 Appendix). Data from each included study will be extracted independently by two reviewers (TOS and KSM). Any disagreements will be addressed through discussion and, if necessary, resolved by a third reviewer (FSVT). The extracted data will encompass details such as the study authors, year of publication, study design, primary objectives, participant characteristics (including inclusion criteria, mean age, sex, type of cancer), description of the interventions, outcome measurement scales, variables related to intervention, and outcome measures, as pain, anxiety and quality of life.

## Missing data

If essential data are missing, the corresponding authors or co-authors of the relevant studies will be contacted via telephone or email to request the necessary information. Should the authors fail to respond or provide the data, the missing information will be excluded from the analysis and its absence will be addressed in the discussion section.

## Analysis

The R Software V.4.3.1 will be used to enter data. The Odds Ratio (OR) and 95% CI for each research will be extracted or computed for dichotomous data. The Mean Difference (MD) or Standardized Mean Difference (SMD) will be computed for continuous data.

The studies will be combined using the random-effects model in the event of heterogeneity ($I2 > 50\%$), and the DerSimonian-Laird method will be used to get the OR and 95% CI. The robustness of the findings in relation to study quality and sample size will be investigated using sensitivity analysis. Only if a meta-analysis is successful will this be feasible. In a summary table, the sensitivity analysis will be shown.

However, we will conduct a subgroup analysis for the different digital Health Interventions. A two-sided p-value <0.05 will be considered statistically significant. If it is not possible to perform a meta-analysis, a qualitative analysis will be carried out.

## Bias risk assessment

The bias risk of the selected articles will be assessed independently by two reviewers using the updated Cochrane risk-of-bias tool for randomized trials (RoB2) [25]. For non-randomized studies, the Risk of Bias in Non-Randomized Studies of Interventions (ROBINS-I) [26] tool will be applied to evaluate potential bias in the included interventions.

The specific criteria that will be evaluated include random sequence generation to assess selection bias, allocation concealment to further evaluate selection bias, blinding of participants and personnel to address performance bias, analysis of incomplete outcome data to determine attrition bias, examination of selective reporting to identify reporting bias, and the identification of any other potential sources of bias. In case of insufficient details reported in the study, authors will be contacted. Discrepancies about the assessment of risk of bias will be solved through discussion with a third author (FSVT).

Whenever possible, funnel plots will be used to assess the potential existence of publication bias, complemented by Egger's [27] weighted correlation and Begg's regression intercept at a 5% significance level.

## Grading quality of evidence

The Grading of Recommendations Assessment, Development, and Evaluation (GRADE) [28] approach will be employed to assess the overall certainty of evidence derived from the included studies. This assessment will consider key domains such as risk of bias, consistency, directness, and precision. The results will inform the overall quality rating of the evidence. The GRADE framework categorizes the quality of evidence as high, moderate, low, or very low. In cases where information is insufficient to complete the assessment, study authors will be contacted via email. Two reviewers will independently perform the quality assessment, and any disagreements will be resolved through discussion (FSVT).

## Discussion

This protocol aims to synthesize evidence on how technologies, such as remote monitoring devices, mobile applications, and digital platforms, can support the management of signs and symptoms in adult oncology patients receiving palliative care. The goal is to compare the effectiveness of remote care versus outpatient care in managing cancer-related symptoms in this population.

Telemedicine has emerged as an innovative and effective strategy for pain management in cancer patients receiving palliative care [29] especially in a global context where there's an increasing need for accessible and continuous healthcare.

Over the past ten years, the use of digital health in palliative care has grown significantly, establishing itself as an emerging and expanding industry. This field combines healthcare services, information technology, and mobile technology, transforming the traditional approach and reducing the need for in-person consultations in outpatient settings [30].

Among the most common signs and symptoms in adult cancer patients receiving palliative care, pain and fatigue stand out as the most troublesome. Additionally, emotional issues and, in some cases, episodes of dyspnea have also been observed, contributing to a significant reduction in the quality of life of these patients [31].

Concerned with the well-being of this patient group, it is essential to develop strategies aimed at improving quality of life, prioritizing therapies that help in the effective management of signs and symptoms. These interventions should minimize the physical effort of patients and reduce their exposure to environments that may pose additional risks, promoting comprehensive and humanized care.

Pain is a predominant symptom in various diseases, especially in cancer patients, and is one of the main causes of suffering for both the patient and their family. It interferes with the ability to carry out simple daily activities, contributing to the emergence of other symptoms, such as depression and a decrease in the desire for self-care [32].

Digital health is an area that offers numerous opportunities to foster innovation and improve the delivery of medical care. Its goal is to make healthcare services more effective, accessible, and efficient by using technology to collect, analyze, store, and share health information [33].

In Brazil, a study published in 2021 analyzed 1,645 teleconsultations conducted between April 2020 and February of the following year, involving 470 cancer patients receiving palliative care [7]. The research identified pain as the most frequent complaint, present in 32.7% of the consultations. The results demonstrated the effectiveness of telemedicine in symptom monitoring, allowing patients and caregivers to stay at home, reducing the need for travel and the risk of exposure to other opportunistic diseases.

In line with the previously mentioned study, another study presented at the ASCO (American Society of Clinical Oncology) in 2024 evaluated 1,250 patients with advanced lung cancer, comparing in-person consultations and telemedicine consultations.

The results showed that the group receiving teleconsultations had a better quality of life, as well as lower levels of anxiety and depression, compared to the in-person group. Patient satisfaction was similar between both groups, highlighting telemedicine as an effective and viable alternative for oncological palliative care [34].

In summary, telemedicine has proven to be an effective tool for pain management and monitoring oncological patients in palliative care, providing a safe and viable alternative for continued care, especially in contexts with restrictions on in-person visits. This strategy stands out as an efficient approach aimed at improving the quality of care provided.

## Supporting information

**S1 File. Peer Review of Electronic Search Strategies.**
(DOCX)

**S2 File. PRISMA-P 2015 Checklist.**
(DOCX)

**S1 Appendix. Categorization of selected studies.**
(XLSX)

## Author contributions

**Conceptualization:** Thiago Oliveira Dos Santos, Fábila Fernanda dos Passos da Rosa, Kleyton Santos de Medeiros, Francis Solange Vieira Tourinho.

**Methodology:** Thiago Oliveira Dos Santos, Fábila Fernanda dos Passos da Rosa, Kleyton Santos de Medeiros.

**Project administration:** Kleyton Santos de Medeiros, Francis Solange Vieira Tourinho.

**Supervision:** Kleyton Santos de Medeiros, Francis Solange Vieira Tourinho.

**Validation:** Thiago Oliveira Dos Santos, Kleyton Santos de Medeiros, Francis Solange Vieira Tourinho.

**Visualization:** Thiago Oliveira Dos Santos, Fábila Fernanda dos Passos da Rosa, Kleyton Santos de Medeiros, Francis Solange Vieira Tourinho.

**Writing – original draft:** Thiago Oliveira Dos Santos, Fábila Fernanda dos Passos da Rosa, Kleyton Santos de Medeiros, Francis Solange Vieira Tourinho.

**Writing – review & editing:** Kleyton Santos de Medeiros, Francis Solange Vieira Tourinho.

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
