## [Editor Report · Decision Letter 0]

6 Aug 2025

PONE-D-25-27115Impact of Digital Health Interventions on Pain and Symptom Management in Home Hospice Patients: A Systematic Review and Meta-Analysis ProtocolPLOS ONE

Dear Dr. Tourinho,

Thank you for submitting your manuscript to PLOS ONE. After careful consideration, we feel that it has merit but does not fully meet PLOS ONE’s publication criteria as it currently stands. Therefore, we invite you to submit a revised version of the manuscript that addresses the points raised during the review process.

We look forward to receiving your revised manuscript.

Kind regards,

Giovanni Ottoboni, Psy, PhD

Academic Editor

PLOS ONE

Journal Requirements:

2. Please include a caption for figure 1.

Additional Editor Comments (if provided):

The authors state that this protocol has been registered in the International Prospective Register of Systematic Reviews (PROSPERO). Could the authors kindly clarify the rationale for presenting the same material here, and explain how this submission adds value beyond the registered protocol?

---

## [Author Response · Author response to Decision Letter 1]

13 Sep 2025

Manuscript ID: PONE-D-25-27115

Title: Impact of Digital Health Interventions on Pain and Symptom Management in Home Hospice Patients: A Systematic Review and Meta-Analysis Protocol

Corresponding Author: Dr. Tourinho

Response to Reviewers’ Comments

We would like to thank the Editors and Reviewers for their thoughtful and constructive comments. We have addressed each point below. All changes are indicated in the tracked version of the revised manuscript and are described below in a point-by-point response.________________________________________

EDITORIAL COMMENTS

1. “Please ensure that your manuscript meets PLOS ONE's style requirements, including those for file naming. The PLOS ONE style templates can be found ...

Response: Thank you for your clarification. We have revised the manuscript, including the abstract structure, in accordance with the journal’s guidelines and references.

de Oliveira Monteiro AB, Honnef LR, Dias de Oliveira JM, Pauletto P, Massignan C, Stefani CM, Zimmermann GS, Canto GL. Prevalence of periodontitis in adolescents: A systematic review protocol. PLoS One. 2025 May 5;20(5):e0321993. doi: 10.1371/journal.pone.0321993. PMID: 40324012; PMCID: PMC12052176.

Fosu PK, Hoor GT, Adjei CA, Atibila F, Ruiter RAC. Prevalence of Hepatitis C viral infection in Ghana: A systematic review and meta-analysis protocol. PLoS One. 2025 Apr 16;20(4):e0321483. doi: 10.1371/journal.pone.0321483. PMID: 40238759; PMCID: PMC12002434.

2. Please include a caption for figure 1.

Response: The figure caption has been included.

Response: Understood; however, no specific reference was recommended for citation.

Additional Editor Comments (if provided):

The authors state that this protocol has been registered in the International Prospective Register of Systematic Reviews (PROSPERO). Could the authors kindly clarify the rationale for presenting the same material here, and explain how this submission adds value beyond the registered protocol?

Response: We are grateful for the opportunity to clarify this important point. Indeed, as indicated, this protocol was previously registered in the International Prospective Register of Systematic Reviews (PROSPERO), in accordance with international best practices for transparency and bias reduction in systematic reviews.

The submission of the protocol to this journal aims to make it more widely accessible and provide a more detailed account than is possible through PROSPERO, as well as to enable peer review, which is not available on the PROSPERO platform. As emphasized in the Cochrane Handbook for Systematic Reviews of Interventions (version 6.4, section 2), the publication of protocols in peer-reviewed journals is a key step toward promoting transparency, increasing the visibility of the study, and preventing unnecessary duplication of efforts.

"Registering and publishing a protocol can help avoid unplanned duplication and enable readers to compare what was planned in the protocol with what was actually done in the review." (Cochrane Handbook, Section 2)

While PROSPERO registration requires only a brief description of the planned methods, the manuscript submitted to the journal allows for a more comprehensive presentation of the rationale, inclusion and exclusion criteria, search strategies, methods of analysis, and assessment of the quality of evidence. Such detailed reporting is not possible within the PROSPERO registry, yet it is essential to ensure reproducibility and enable critical appraisal by the scientific community.

Moreover, the formal publication of the protocol contributes to scientific integrity by providing a citable and indexed record, thereby strengthening the methodological traceability of the forthcoming systematic review. This process is widely recognized as good practice and is encouraged by initiatives such as PRISMA-P (Preferred Reporting Items for Systematic Review and Meta-Analysis Protocols), which emphasizes the importance of thoroughly describing methods prior to conducting the review.

We hope this explanation meets the editorial request and remain at your disposal for any further clarification.

Sincerely,

Dr. Francis Tourinho, on behalf of all co-authors

---

## [Editor Report · Decision Letter 1]

15 Sep 2025

Impact of Digital Health Interventions on Pain and Symptom Management in Home Hospice Patients: A Systematic Review and Meta-Analysis Protocol

PONE-D-25-27115R1

Dear Dr. Tourinho,

We’re pleased to inform you that your manuscript has been judged scientifically suitable for publication and will be formally accepted for publication once it meets all outstanding technical requirements.

Kind regards,

Giovanni Ottoboni, Psy, PhD

Academic Editor

PLOS ONE
---

## [Editor Report · Acceptance letter]

PONE-D-25-27115R1

PLOS ONE

Dear Dr. Tourinho,

I'm pleased to inform you that your manuscript has been deemed suitable for publication in PLOS ONE. Congratulations! Your manuscript is now being handed over to our production team.

Kind regards,

on behalf of

Professor Giovanni Ottoboni

Academic Editor

PLOS ONE